# Dental Treatment Needs and Related Risk Factors among School Children with Special Needs in Taiwan

**DOI:** 10.3390/jpm11060452

**Published:** 2021-05-23

**Authors:** Szu-Yu Hsiao, Ping-Ho Chen, Shan-Shan Huang, Cheng-Wei Yen, Shun-Te Huang, Shu-Yuan Yin, Hsiu-Yueh Liu

**Affiliations:** 1Division of Pediatric Dentistry and Special Needs Dentistry, Department of Dentistry, Kaohsiung Medical University Hospital, Kaohsiung 807377, Taiwan; szyuhs@kmu.edu.tw (S.-Y.H.); 1070480@kmuh.org.tw (S.-S.H.); 1040474@kmuh.org.tw (C.-W.Y.); shunteh@kmu.edu.tw (S.-T.H.); 2School of Dentistry, College of Dental Medicine, Kaohsiung Medical University, Kaohsiung 807378, Taiwan; phchen@kmu.edu.tw; 3Department of Oral Hygiene, College of Dental Medicine, Kaohsiung Medical University, Kaohsiung 807378, Taiwan; 4Department of Nursing, Fooyin University, Kaohsiung 831301, Taiwan; EN006@fy.edu.tw; 5Department of Medical Research, Kaohsiung Medical University Hospital, Kaohsiung Medical University, Kaohsiung 807377, Taiwan

**Keywords:** disability, children, oral health, caries, dental treatment

## Abstract

The purpose of this study was to assess dental treatment needs (TNs) and related risk factors of children with disabilities (CD). This cross-sectional study recruited 484 CD, 6 to 12 years of age, from 10 special education schools in Taiwan. Dental status and TNs were examined and evaluated by well-trained dentists and based on the criteria set by the World Health Organization (1997). The results indicated that 61.78% required restorative dental treatment due to their dental caries. On average, each participant had 2.72 teeth that required treatment, and 6.38 surfaces required restoration. One-quarter of the participants (24.79%) required 1- or 2-surface restoration, and one out of three (36.98%) had more complex TNs (including 3 or more surfaces to be filled, pulp care, extraction, and more specialized care). The significant risk factors associated with restorative TNs among CD were those whose parents had lower socioeconomic status, frequent sweets intake, insufficient tooth-brushing ability, and poor oral health. Most of the CD had extensive unmet TNs for their caries and required complex treatment to recover the function of their teeth. Encouraging parents/caregivers to take their children for dental treatment, promoting awareness of the importance of dental hygiene, giving assistance to brushing their teeth after eating, and controlling and/or modifying sweet diet habits are necessary to reduce CD’s dental caries, especially those with lower socioeconomic status parents/caregivers.

## 1. Introduction

People with disabilities usually suffer from a significantly higher prevalence of poor dental hygiene, plaque accumulation, dental caries, gingivitis, and periodontal disease than the ordinary population, and it gets worse with increasing age [1,2,3,4,5,6]. The top three medical care departments most frequently visited by people with intelligence disabilities in Taiwan are internal medicine (24.4%), psychiatry (16.7%), and dentistry (13.8%), as opposed to surgery, family medicine, and obstetrics and gynecology, which are utilized mainly by the general population [7]. People with disabilities have higher dental treatment needs (TNs) caused by secondary conditions as opposed to the general population, who seek and receive healthcare services regularly. According to Healthy People 2010 in the USA, secondary conditions refer to the problems related to medical, social, emotional, family, or community problems that a person with a primary disabling condition may encounter in his/her life [8]. It often aggravates and/or lessens their life quality in terms of physical, psychosocial, and social functions and increases the burden of health care on their parents, family members, or other caregivers with limited resources. Koritsas & Iacono reported that people with disabilities experienced an average of 11.3 secondary conditions (including medical complications, psychiatric disorders, environmental and quality-of-life issues, and difficulties with access to medical care/centers) during their development [9]. Secondary conditions often cause significant limitations, including reading difficulties, communication, physical fitness–conditioning, personal hygiene–appearance, weight, dental and oral hygiene, and memory problems [9]. Dental and oral hygiene is one of the significant limitations which is caused by secondary conditions.

A previous study reported that the greatest unmet health needs of people with disabilities are unmet dental treatment [10]. Studies have shown that children with disabilities (CD) usually receive less restorative care compared to the general population [11,12,13,14,15]. People with disabilities cannot express their physical discomfort properly because of their physical, intellectual, and psychological barriers. Long-term neglect of TNs, with increasing severity and complexity of oral diseases, leads to delay of time-sensitive, necessary treatment [16]. A previous study reported that only 32.37% of decayed teeth received restorative treatment among 6- to 12-year-old CD, and it was significantly lower than the ordinary population (47.72%) in Taiwan [17,18,19]. It reveals that the dental TNs for those children are unmet.

Dental treatment is a basic component of rehabilitation for CD, and it is difficult for dentists to perform [20]. The majority of children with special needs can be adequately treated using non-pharmacologic behavior management such as the tell-show-do technique [14,21]. Disabled people who have extensive and severe dental problems very often cannot cooperate well during the dental treatment process, and the treatment has to be performed by pharmacological behavior management techniques such as nitrous oxide/oxygen sedation, oral sedation. or general anesthesia (GA) to achieve higher quality treatment [14,22,23]. In order to avoid oral diseases’ worsening and to provide CD with effective dental treatment services, it is helpful to provide appropriate baseline information regarding trends in unmet needs and related risk factors so as to make proper decisions to improve the oral health status of CD. Thus, the present study was carried out in an attempt to determine the unmet dental TNs and related factors of children with various disabilities in primary schools for CD in Taiwan.

## 2. Method

### 2.1. Study Design and Sample Characteristics

This cross-sectional study recruited 484 children, aged from 6 to 12 years old (mean age = 9.47 ± 2.06 years old), from 10 out of 18 special education primary schools in Taiwan. According to the definition of the Physically and Mentally Disabled Citizens Protection Act [24], people with disabilities refers to those who are limited or restricted from engagement in ordinary living activities and participation in society. All the participants have been evaluated by a committee composed of professionals from medicine, social work, special education, and employment counseling. After evaluation, they receive certificates that prove their classification and severity of disabilities after the processes of evaluation and assessment.

The CD were categorized according to their disability certificates in this study, which included vision disability (VD); hearing, voice or speech mechanism disability (HVD); intellectual disability (ID); and multiple disabilities (MD). MD means at least two or more types of impairment. The participants of this study included the certified MD children with ID and at least one or more co-occurring conditions. These conditions included vision disability, hearing mechanism disability, limb disability, etc.

### 2.2. Ethical Approval

This study was conducted according to the guidelines of the Declaration of Helsinki and approved by the Institutional Review Board of Kaohsiung Medical University Hospital (Protocol number: KMUH-IRB-950125). The study purpose, procedures, and contents of the survey were explained to the parents or guardians of all CD, and written consent was obtained from parents or guardians who agreed to allow their children to participate.

### 2.3. Data Collection

#### 2.3.1. Oral Examination

Examination of the oral health status of study participants was carried out using a disposable dental mirror (Prosperity Island Medical Dressing Co, Ltd., Changhua, Taiwan), a standard dental explorer (CP-11.5B6, Hu-Freidy, Chicago, IL, USA), and a flashlight, and with the help of nursing staff in the classroom, lobby, auditorium, or other open space of the schools. An oral examination of the oral health status of study participants was performed in accordance with the methods and criteria of the World Health Organization [25], carried out by six well-trained dentists (S.-T.H., S.-Y.H., C.-Y.J., B.-M.C., C.-C.C.,R.-C.T.) who were evaluated prior to the survey by a senior dentist (gold standard: S.T.H.) with abundant clinical experience, and achieved acceptable reliability and inter-examiner agreement with a Kappa score between 0.80 and 0.84 (0.81, 0.84, 0.80, 0.80, and 0.82) between each two dentists.

The oral examination and records were determined on the basis of a maximum of 20–28 teeth for primary, mixed, or permanent dentition up to 12 years old. Unmet dental TNs were evaluated using dmt + DMT and dms + DMS indices to record numerically the number of unmet dental treatment teeth and surfaces. The dmt + DMT index is a sum of decayed teeth number (dt + DT) and missing teeth number (mt + MT) of mixed dentition. The dms + DMS index is a sum of decayed teeth surfaces (ds + DS) and missing surfaces (dm + DM) of mixed dentition. These two indices provide the consequences of untreated caries and present an aggregate value of current dental TNs.

Plaque and gingivitis indices were evaluated on 6 indicative surfaces (buccal side of 4 teeth and lingual side of 2 teeth) with the naked eye and a CPI explorer (CP-11.5B6, Hu-Freidy, Chicago, IL, USA). The 4 buccal surfaces selected for inspection were from 2 posterior teeth (tooth numbers 55 or 17/16 and 65 or 26/27) and 2 anterior teeth (tooth numbers 51 or 11 and 71 or 31). The 2 lingual surfaces selected for inspection were from 2 posterior teeth of the mandible (tooth numbers 85 or 47/46 and 75 or 36/37). The value of the poorest surface situation prevailed.

Plaque index (PI) was assessed according to criteria modified from Greene and Vermillion [26] and marked as: 0 = no plaque; 1 = plaque covering not more than one-third of the exposed tooth surface; 2 = plaque covering more than one-third, but not more than two-thirds, of the exposed tooth surface; and 3 = plaque covering more than two-thirds of the exposed tooth surface. The tooth surface with the worst plaque of six teeth was recorded as a whole-mouth oral hygiene score for each participant. Gingivitis index (GI) was assessed by inspection of the following and tendency toward spontaneously bleeding criteria: 0 = healthy; 1 = mild gingivitis—no bleeding on probing; 2 = moderate gingivitis—bleeding on probing; 3 = severe gingivitis—ulceration [27]. The most-inflamed gingival surface of six teeth was identified as a whole gingivitis severity score for each participant. The PI and GI were classified into yes (score ≥ 1) and no (score = 0) for statistical analysis, respectively.

The dental TNs of each child were categorized as simple, moderate, and complex based on the following criteria [12,28]: simple = child had no caries and required no restorative treatment, but required assistance with oral hygiene and preventive treatment such as oral hygiene instruction (OHI), scaling, application of topical fluoride, and/or fissure sealants; moderate = one or more teeth caries and required one and/or two surface restorations; complex = one or more teeth caries and required three or four surface restorations/stainless steel crowns, endodontic therapy/crown, extraction, and/or prosthodontics.

#### 2.3.2. Questionnaire

The self-report questionnaire was completed by the parents or guardians of the participants. The questionnaire consisted of close-ended items and was constructed in three parts: demographic data and family characteristics, sweet intake habits, and tooth-brushing habits of the participant. Sweet foods were considered as highly fermentable carbohydrates made with rich sugar to add a sweet taste such as chocolate, candy, cake, baked goods, ice cream, carbonated beverages, juice, milk with sugar, and chewing gum with sugar. The definition of the classification of parents’ educational and occupation levels referred to the publications of Liu et al. [18]. The parents’ educational levels were classified as follows: both low, one low and one high, or both high. The parents’ occupation levels were divided into both unskilled, one unskilled and one skilled, or both skilled.

### 2.4. Statistical Analysis

All data were analyzed with JMP version 14 (SAS Institute, Cary, NC, USA). Categorical variables in a group were compared using Pearson’s χ^2^ test and Fisher’s exact test and were presented as frequency and percentage, and the differences between numerical variables were analyzed using a *t*-test and one-way analysis of variance (ANOVA) and presented as the mean and standard deviation (SD). Both univariate and multivariate logistic regression models were estimated to assess the unadjusted and adjusted associations of a preset independent variable with TNs of dental caries. Only the independent variable, as the risk factor that was found to be significantly associated with dental TNs in the univariate logistic regression, was included in the multiple logistic regression models. A significance level of the *p*-value was set at 5%.

## 3. Results

The younger children had statistically significantly higher TNs of decayed teeth and surfaces than the older ones (all *p* < 0.001). We found that children with HVD had the highest TNs of teeth (3.48 ± 3.74) and surfaces (8.89 ± 11.57) compared to children with VD (1.87 ± 2.87 and 4.14 ± 8.59) (*p* = 0.023 and *p* = 0.030). The children with mild/moderate disabilities or with HVD had a statistically significantly higher decayed teeth and surfaces need for dental treatment than those with the other severities of disabilities or classifications of disabilities (*p* = 0.017 and *p* = 0.016). TN of surfaces statistically significantly decreased when the parents had higher educations or the parents’ occupations tended to be skilled (*p* = 0.023 and *p* = 0.018). The results also showed the CD who asked for sweets, frequently had sweets, had independent tooth-brushing abilities, or infrequently brushed their teeth every day tended to have statistically significantly higher decayed teeth and surfaces for treatment (all *p* < 0.05) (Table 1).

Overall, one-fourth of the children needed moderate treatment for one or two surface fillings, and more than one-third of the children needed complex treatment such as three or more surface fillings, pulp care, or crowns (Table 2). Dental treatment modalities had a statistically significantly positive association with age, severity of disability, parents’ educational and occupation levels, asking for sweets, frequency of sweets intake, sweets as a reward for behavior control, and tooth-brushing ability of the participants (all *p* < 0.05). The dental TN modality of children with mild/moderate disability was statistically significantly more complex than those with severe/profound disability (*p* = 0.009). The higher the educational or occupational level of the parents, the less and simpler the dental TN modalities of their children (*p* = 0.015 and *p* = 0.045). Dental treatment modalities had a statistically significant positive association with the behaviors of asking for sweets and frequency of eating sweets (*p* = 0.021 and *p* = 0.001) among CD.

In order to better understand the participants, we further compared the oral status, demographics, sweet intake, and daily tooth-brushing habits among different disabilities, as shown in Table 3. We found the VD children had the lowest proportion of plaque and gingivitis (*p* = 0.036 and *p* < 0.001, respectively) than other disabled children. The MD children asked for sweets less, and their daily tooth-brushing is hard to perform by themselves. Less of the VD children’s behavior was controlled by giving sweets as a reward than the others. Compared with other disabled children, the HVD children had a statistically significantly lower rate (8.24%) of brushing their teeth three or more-than-three times a day (*p* < 0.001).

Multiple logistic regression models showed the major risk factors for restorative TNs among CD were their intake of sweets at least once a week (AOR = 2.45, 95% CI = 1.48–4.11, *p* = 0.001) or more frequently than those who never or sometimes consumed sweets. The other risk factors—significantly related to children’s poor oral health were having gingivitis (AOR = 1.94, 95% CI = 1.22–3.11, *p* = 0.006) and insufficient ability to brush their teeth without assistance from parents/caregivers (AOR = 1.95, 95% CI = 1.19–3.22, *p* = 0.008) (Table 4).

## 4. Discussion

The main findings of this study identified a higher caries prevalence and extensive unmet dental TNs among CD in special schools in Taiwan. High frequency of sweet intake and inadequate tooth-brushing ability were the critical risk factors attributed to CD’ needs for extensive restorative care. Even though some CD have better independent abilities, they have the potential need for attention and assistance from their parents and/or caregivers to maintain and/or protect their oral hygiene with daily tooth-brushing behavior. This is beneficial to reduce the high dental caries treatment needed for their teeth and teeth surfaces.

The oral hygiene of CD is poor. This can be confirmed from our results that nearly 80% of CD had a mild-to-severe plaque index, and approximately 50% of them had gingivitis. Due to these CD being exposed to such a poor oral environment, 61.78% of them developed dental caries. If plaque is not completely removed from the teeth surfaces every day, a mild form of periodontal disease, which is called gingivitis, occurs. In our study, 47.93% of CD had inflammation and bleeding of the gums. This results make us more convinced that poor oral hygiene will affect subsequent oral diseases such as dental decay and gingivitis, which is consistent with previous findings [12,18,28]. However, good oral hygiene can be achieved via regular removal of plaque by thorough tooth-brushing every day. If the plaque declines, tooth decay and gingivitis will simultaneously diminish.

Dental caries remains a primary health problem among CD. Unmet decayed and missing teeth in 6–12-year-old Taiwanese CD, with a mean age of 9.47 years old, have decreased, with the mean caries experience index declining from 3.36 teeth (72.77%) in 2005 [17] to 2.72 teeth (61.78%) in our study (2021). The same index in 9-year-old children also declined from 2.70 (73.55%) in 2007 to 2.66 (47.09%) in 2012 [17,29] among the ordinary population. The decreasing value of caries experience index among CD (10.99%) in our study was less than in those without disabilities (26.46%) [17,29]. In comparison with other studies, the caries prevalence in this study (61.78%) is higher than the result in Australia (56%) but lower than that in Iran (73.6%) [12,28]. The studies of 5–16-year-old CD in Australia and Iran reported less unmet dental treatment teeth (1.53 and 2.09), which was half to two-thirds of our findings (2.75) [12,28]. Furthermore, we compared the treatment modalities and found the CD in the present study had a higher need of complex treatment (36.64%) than the CD in the Australian (21%) and Iranian (25.1%) studies [12,28]. Fewer CD had caries lesions in the present study than in Iran, but they needed more complex dental treatment. It appeared that the CD in our study tended to have more severe tooth caries and required higher dental TNs than CD from other countries [28].

The moderate and complex dental TNs among CD were unmet in this study. Most of the patients receiving dental treatment under GA were people with special needs [14,21,30], especially when the patient needed to have extensive and complicated treatment [31]. The most common treatments (extraction, restoration, and pulp therapy) under GA over the past 10 years in Taiwan are related to a high proportion of multiple dental caries (86.4%) [32,33]. Our result is consistent with other studies, wherein children who need to be treated under GA usually have a higher unmet decayed teeth treatment of more than 10 teeth [32,33]. There were two-thirds (59.87%) out of 36.98% of our participants who might need to be treated under GA due to their uncooperative behavior in specialist clinics or regional hospitals that have critical care facilities. If the CD receive their first dental treatment as early as possible, they can achieve more effective dental rehabilitation [32].

There was a clear negative correlation between decayed teeth and surfaces for complex restorative TNs and severity of disability. This observation is in line with previous studies that children with profound disabilities acquire partial or complete assistance from others as it pertains to maintaining their oral health due to lack of adequate manual dexterity ability [12,22]. However, this is not always true, according to our findings. Our results showed the complex restorative TNs in HVD (71.26%) were higher than MD (58.02%) and VD (52.86%), which was also contrary to the results of a study in Saudi Arabia [34]. The studies of 5–16-year-old children with HVD in Saudi Arabia had restorative TN of 66.02%, which was lower than MD and VD (76.93% and 74.29%) in the present study. In this study, the HVD and VD children had the better capability to learn oral hygiene skills and take care of themselves in comparison with other groups. However, HVD children had ordinal vision and tended to be more susceptible to the allure of various sweets than VD children. The HVD children also had a better independent ability to access, buy, and obtain sweets by themselves than other disability groups. If HVD children do not perform the positive behavior of cleaning their teeth after sweets under supervision by parents/caregivers, it is likely to result in the repercussions of increasing the risk of more caries teeth and surfaces and higher restorative TNs.

Obstacles blocking these participants from achieving lower TNs in this study are improper sweet habits and inadequate tooth-brushing behavior. Manual dexterity is not a guarantee of better oral health and lower TNs among CD. The effectiveness of brushing is limited if CD receive less brushing assistance or supervision when they brush their teeth [18,34,35,36]. The higher the education level the parents/caregivers had, the more aware they were about how to sufficiently complete the oral care needs of the CD [18,36,37]. If parents have higher occupation levels, they will have the sufficient financial capacity to take care of their children’s oral hygiene to reach better oral health and decrease dental TNs [36,37]. However, the negative effect of caries TNs among CD from frequent sweets intake is superior to that from insufficient tooth-brushing ability [38]. The results of our study showed that frequent tooth-brushing could minimize the severity and prevalence of TN. The consumption of sweets in small amounts, along with other fermentable carbohydrates consumed frequently. will increase the caries risk and TNs, rather than large amounts eaten occasionally. Sweet limits, not only controlling the frequency, but also advising to give healthy or low-carcinogenic alternative foods such as fresh fruit and vegetables and/or low-carcinogenic foods by parents/caregivers contributed to a lower prevalence of untreated dental caries and TN rates among CD [39].

Before treatment, how to prevent and decrease the number and severity of TNs is important. Our study showed that tooth-brushing can, in part, diminish the association between giving sweets as a reward in behavior control, asking for sweets, and frequency of sweets intake on dental decay outcome in CDs [38]. Sticky foods with sugar and/or fermentable carbohydrates can stay in the oral environment for longer periods, thus increasing the potential and risk for tooth decay. Similar to ordinary children, having their children brush their teeth at least twice a day is deeply associated with parents’/caregivers’ self-efficacy or confidence [40]. More frequent tooth-brushing might compensate for the inadequate tooth-brushing ability of children, as seen in previous studies [18]. To address the high TNs among CD in the present study, we need to provide promotion courses to encourage the parents/caregivers with lower educational levels, especially those in unskilled occupation levels, to implement effective preventive measures such as brushing after eating and correctly choosing healthy snacks for their children.

There are several limitations to the present study. One is that we have difficulty concluding regarding the causation between dental TNs and related risk factors by self-reported questionnaires, as is the case in most observational cross-sectional studies. Second, the observations were assessed outside the dental clinic, with limited accessibility and difficulties for patients’ handling. Third, a small number of parents/caregivers might answer questions in such a way as to meet social expectations and thus cause answer bias. In addition, we collected the data from special schools, not including bedridden and homebound groups; therefore, our results may underestimate the real consideration of all disability groups and reflect the status of school CD. However, our findings may help to better understand the TNs of CD and propose some measures to improve their oral care.

## 5. Conclusions

The present findings illustrate TNs with an enormous need for restorative treatment, concentrated on 61.78% of CD. Improper sweet habits and inadequate tooth-brushing ability were found in the risk factors for having the highest odds of requiring restorative treatment. Study results reveal that oral problems among these children do not get much attention from their parents/caregivers, especially those with lower education or occupation levels. It is imperative to encourage low-educational-level parents/caregivers to take their children for dental treatment, teach them why and how to brush teeth after eating, and help them modify the sweet intake habits of their children to decrease their children’s caries and improve their children’s oral health.

## Figures and Tables

**Table 1 jpm-11-00452-t001:** Dental treatment needs of teeth and surfaces by demographics and dietary and tooth-brushing habits among children with disabilities.

Variable	*N*	Teeth for Treatment Need		*p*-Value	Surfaces for Treatment Need		*p*-Value
		Mean	(SD)		Mean	(SD)	
Total	484	2.72	(3.64)		6.38	(10.50)	
Age							
6–7 years old	145	4.23	(4.71)	<0.001	10.32	(14.21)	<0.001
8–9 years old	126	2.86	(3.48)		6.41	(9.72)	
10–12 years old	213	1.71	(2.33)		3.69	(6.37)	
Gender							
Male	297	2.64	(3.49)	0.330	6.09	(10.35)	0.439
Female	187	2.97	(3.87)		6.85	(10.74)	
Severity of disability							
Mild/moderate	111	3.51	(4.02)	0.023	8.69	(12.10)	0.018
Severe/profound	373	2.54	(3.50)		5.70	(9.89)	
Classification of disability							
Vision disability	70	1.87	(2.87)	0.023	4.14	(8.59)	0.030
Hearing, voice, or speech disability	87	3.48	(3.74)		8.89	(11.57)	
Intelligence disability	115	2.82	(3.58)		6.78	(10.69)	
Multiple disabilities	212	2.74	(3.82)		5.88	(10.37)	
Parents’ educational level							
Both low/one low, one high	250	2.96	(3.59)	0.039	7.22	(11.01)	0.017
Both high	200	2.27	(3.34)		4.88	(9.41)	
Parents’ occupation							
Both unskilled	196	2.98	(3.53)	0.065	7.17	(10.64)	0.016
One unskilled, one skilled	129	2.49	(3.15)		5.09	(8.30)	
Both skilled	80	1.98	(3.06)		3.98	(7.56)	
Ask for sweets							
No	304	2.32	(3.17)	0.001	5.12	(9.08)	0.001
Yes	169	3.54	(4.26)		8.59	(12.21)	
Frequency of sweets intake							
Never/sometimes	116	1.56	(2.29)	<0.001	3.07	(6.15)	<0.001
At least once a week	221	3.22	(4.11)		7.43	(11.65)	
At least once a day	118	3.19	(3.60)		7.69	(10.35)	
Sweets as a reward in behavior control							
No	283	2.25	(3.05)	0.002	4.83	(8.66)	0.001
Yes	171	3.41	(4.10)		8.24	(11.79)	
Appetite							
Good	380	2.54	(3.46)	0.020	5.65	(9.93)	0.009
Poor	104	3.59	(4.15)		9.07	(12.06)	
Tooth-brushing ability							
Independent	308	3.10	(3.83)	0.004	7.22	(11.03)	0.012
Dependent	155	2.12	(3.13)		4.79	(9.13)	
Frequency of tooth-brushing each day							
Sometimes	15	4.73	(5.87)	0.036	12.60	(17.82)	0.017
1–2 times	353	2.84	(3.59)		6.63	(10.49)	
≥3 times, after meals	95	2.22	(3.27)		4.62	(8.48)	

**Table 2 jpm-11-00452-t002:** Dental treatment modalities by demographics and dietary and tooth-brushing habits among children with disabilities.

Variable	*N*	Simple		Moderate		Complex		*p*-Value
		*n*	(%)	*n*	(%)	*n*	(%)	
Total	484	185	(38.22)	120	(24.79)	179	(36.98)	
Age								
6-7 years old	145	43	(23.24)	34	(28.33)	68	(37.99)	0.025
8-9 years old	126	48	(25.95)	32	(26.67)	46	(25.70)	
10-12 years old	213	94	(50.81)	54	(45.00)	65	(36.31)	
Gender								
Male	297	118	(63.78)	74	(61.67)	105	(58.66)	0.602
Female	187	67	(36.22)	46	(38.33)	74	(41.34)	
Severity of disability								
Mild/moderate	111	31	(16.76)	26	(21.67)	54	(30.17)	0.009
Severe/profound	373	154	(83.24)	94	(78.33)	125	(69.83)	
Classification of disability								
Vision disability	70	33	(17.84)	16	(13.33)	21	(11.73)	0.090
Hearing, voice, or speech disability	87	25	(13.51)	20	(16.67)	42	(23.46)	
Intelligence disability	115	38	(20.54)	30	(25.00)	47	(26.26)	
Multiple disabilities	212	89	(48.11)	54	(45.00)	69	(38.55)	
Parents’ educational level								
Both low/one low, one high	250	83	(47.43)	62	(57.41)	105	(62.87)	0.015
Both high	200	92	(52.57)	46	(26.29)	62	(37.13)	
Parents’ occupation								
Both unskilled	196	62	(40.79)	54	(52.94)	80	(52.98)	0.045
One unskilled, one skilled	129	49	(32.24)	34	(33.33)	46	(30.46)	
Both skilled	80	41	(26.97)	14	(13.73)	25	(16.56)	
Ask for sweets								
No	304	130	(71.43)	73	(63.48)	101	(57.39)	0.021
Yes	169	52	(28.57)	42	(36.52)	75	(42.61)	
Frequency of sweets intake								
Never/sometimes	116	60	(34.29)	27	(23.89)	29	(17.37)	0.001
At least once a week	221	75	(42.86)	64	(56.64)	82	(49.10)	
At least once a day	118	40	(22.86)	22	(19.47)	56	(33.53)	
Sweets as a reward in behavior control								
No	283	120	(68.18)	72	(63.72)	91	(55.15)	0.043
Yes	171	56	(31.82)	41	(36.28)	74	(44.85)	
Appetite								
Good	380	154	(83.24)	93	(77.50)	133	(74.30)	0.110
Poor	104	31	(16.76)	27	(22.50)	46	(25.70)	
Tooth-brushing ability								
Independent	308	104	(58.10)	80	(71.43)	124	(72.09)	0.010
Dependent	155	75	(41.90)	32	(28.57)	48	(27.91)	
Frequency of tooth-brushing each day								
Sometimes	15	5	(2.79)	4	(3.57)	6	(3.49)	0.969
<3 times	353	135	(75.42)	87	(77.68)	131	(76.16)	
≥3 times, after meals	95	39	(21.79)	21	(18.75)	35	(20.35)	

**Table 3 jpm-11-00452-t003:** Demographics, dietary and tooth-brushing habits, and oral hygiene status by classification of disability among children with disabilities.

Variable	VD		HVD		ID		MD		*p*-Value
	*n*	(%)	*n*	(%)	*n*	(%)	*n*	(%)	
Total	70	(13.08)	87	(16.26)	115	(21.50)	212	(42.43)	
Age									
6–7 years old	13	(18.57)	26	(29.89)	34	(29.57)	75	(33.96)	0.256
8–9 years old	25	(35.71)	22	(25.29)	27	(23.48)	52	(24.53)	
10–12 years old	32	(45.71)	39	(44.83)	54	(46.96)	88	(41.51)	
Gender									
Male	39	(55.71)	60	(68.97)	76	(66.09)	122	(57.55)	0.142
Female	31	(44.29)	27	(31.03)	39	(33.91)	90	(42.45)	
Severity of disability									
Mild/moderate	6	(8.57)	27	(31.03)	67	(58.26)	11	(5.19)	<0.001
Severe/profound	64	(91.43)	60	(68.97)	48	(41.74)	201	(94.81)	
Parents’ educational level									
Both low/one low, one high	38	(55.88)	55	(68.75)	72	(69.23)	85	(42.93)	<0.001
Both high	30	(44.12)	25	(31.25)	32	(30.77)	113	(57.07)	
Parents’ occupation									
Both unskilled	16	(29.63)	42	(54.55)	58	(62.37)	80	(44.20)	0.006
One unskilled, one skilled	22	(40.74)	21	(27.27)	24	(25.81)	62	(34.25)	
Both skilled	16	(29.63)	14	(18.18)	11	(11.83)	39	(21.55)	
Ask for sweets									
No	44	(63.77)	47	(54.65)	67	(59.82)	146	(70.87)	0.039
Yes	25	(36.23)	39	(45.35)	45	(40.18)	60	(29.13)	
Frequency of sweets intake									
Never/sometimes	19	(29.23)	15	(17.86)	19	(17.59)	63	(31.82)	0.004
At least once a week	37	(56.92)	46	(54.76)	50	(46.30)	88	(44.44)	
At least once a day	9	(13.85)	23	(27.38)	39	(36.11)	47	(23.74)	
Sweets as a reward in behavior control									
No	47	(73.44)	54	(65.06)	63	(58.88)	119	(59.50)	0.183
Yes	17	(26.56)	29	(34.94)	44	(41.12)	81	(40.50)	
Appetite									
Good	45	(64.29)	70	(80.46)	99	(86.09)	166	(78.30)	0.006
Poor	25	(35.71)	17	(19.54)	16	(13.91)	46	(21.70)	
Tooth-brushing ability									
Independent	67	(97.10)	74	(87.06)	77	(71.30)	90	(44.78)	
Dependent	2	(2.90)	11	(12.94)	31	(28.70)	111	(55.22)	<0.001
Frequency of tooth-brushing each day									
Sometimes	1	(1.45)	1	(1.18)	5	(4.63)	8	(3.98)	<0.001
<3 times	35	(50.72)	77	(90.59)	88	(81.48)	153	(76.12)	
≥3 times, after meals	33	(47.83)	7	(8.24)	15	(13.89)	40	(19.90)	
Plaque									
Yes	48	(69.57)	74	(87.06)	97	(89.81)	168	(79.25)	0.036
No	22	(31.88)	13	(15.29)	18	(16.67)	44	(20.75)	
Gingivitis									
Yes	22	(31.43)	52	(59.77)	68	(59.13)	92	(42.45)	<0.001
No	48	(68.57)	35	(40.23)	47	(40.87)	120	(57.55)	

**Table 4 jpm-11-00452-t004:** Logistic regression models of risk factors associated with restorative treatment needs among children with disabilities.

Variable	COR ^a^	95% CI	*p*-Value	AOR ^b^	95% CI	*p*-Value
		(Lower, Upper)			(Lower, Upper)	
Gender						
Female (vs. Male)	1.18	(0.81, 1.73)	0.390	1.17	(0.73, 1.87)	0.517
Age						
6–7 years old (vs. 10–12 years old)	1.71	(1.13, 2.61)	0.012	2.55	(1.45, 4.58)	0.002
8–9 years old (vs. 10–12 years old)	1.01	(0.66, 1.54)	0.973	1.25	(0.71, 2.19)	0.439
Severity of disability						
Mild/moderate (vs. severe/profound)	1.81	(1.15, 2.92)	0.012			
Classification of disability						
Hearing, voice, or speech disability (vs. vision disability)	1.67	(1.02, 2.82)	0.046			
Intelligence disability (vs. vision disability)	1.34	(0.87, 2.10)	0.191			
Multiple disabilities (vs. vision disability)	0.75	(0.52, 1.09)	0.134			
Parents’ educational level						
Both low/one low, one high (vs. both high)	1.71	(1.17, 2.52)	0.006	1.56	(0.95, 2.59)	0.081
Parents’ occupation						
Both unskilled/one unskilled, one skilled (vs. both skilled)	2.03	(1.24, 3.33)	0.005	1.86	(1.04, 3.36)	0.037
Ask for sweets						
Yes (vs. no)	1.68	(1.13, 2.51)	0.011			
Frequency of sweets intake						
At least once a week (vs. never/sometimes)	1.86	(1.23, 2.82)	0.003	2.45	(1.48, 4.11)	0.001
At least once a day (vs. never/sometimes)	1.90	(0.56, 8.67)	0.339	4.01	(0.97, 21.15)	0.070
Sweets as a reward in behavior control						
Yes (vs. no)	1.51	(1.02, 2.26)	0.041			
Appetite						
Poor (vs. good)	1.60	(1.01, 2.59)	0.047			
Tooth-brushing ability						
Independent (vs. dependent)	1.84	(1.24, 2.73)	0.002	1.95	(1.19, 3.22)	0.008
Gingivitis						
Yes (vs. no)	2.01	(1.39, 2.94)	<0.001	1.94	(1.22, 3.11)	0.006

^a^ COR: crude odds ratio. Data analysis by univariate logistic regression model. Dependent variable was with dental treatment needs. ^b^ AOR: adjusted odds ratio. Data analysis by multiple logistic regression model, adjusted participants’ gender and age. Variables found with statistically significant associations in univariate logistic regression analysis were included in the multiple logistic regression models. Dependent variable was with treatment needs.

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
