# Peer review of "Dental Treatment Needs and Related Risk Factors among School Children with Special Needs in Taiwan"

_jpm, 2021, doi:10.3390/jpm11060452_

Round 1

Reviewer 1 Report

Dear Authors,

The paper aims to analyzed the percentage of caries and oral disease among children with special needs.

the article is very interesting but I suggest a revision.

-line 46 " According to Healthy People 2010 in the USA, "

 this should be a citation? if yes please provide it

line 55-57 please rewrite the sentence it is a little bit confusing.

line 84 please provide the mean age

line 105  what type of mirror, what type of dental explore please provide brand and type for each.

line 111 please add names abbreviation of the six trained  and the "gold standard" (S.S)

line 117 please explain the abbreviation  "evaluated using dmt + DMT and dms + DMS indices to "

line 123 "CPI" add brand and type

line 169 please use the already cited abbreviations.

"The children with mild/moderate dissbilities or with hearing or voice disability had a statistically significantly higher decayed  teeth and surfaces need for dental treatment than those with the other severity of disability 171 or classification of disabilities "

Mild-moderate disabilities (MD)?

hearing or voice disability  (HVD)?

please rewrite the discussion 

There are several repetitions about the same topic (e.g. "proper sweet habits and inadequate tooth-brushing behavior" 

"diet control or food habits play a critical role in the causation of dental caries" 

"Most TNs originate from dental caries, which is associated with oral care habits and CD’s tooth-brushing behavior"

"If parents have higher occupation level, we inferred that the parents 295 will have sufficient financial capacity to take care of their children's oral health "

plase check the discussion too long and repetitive.

Iin the limitations, you should add that the observation was assessed outside the dental clinic with limited accessibility and difficulties for the patients handling.

Author Response

Author's Response to Decision Letter for (Manuscript ID: jpm-1206711)

Title: Dental treatment needs and related risk factors among school children with special needs in Taiwan

Dear Reviewer,

The authors sincerely appreciate for the opportunity to consider our manuscript for publication, subject to major amendments. We have revised this manuscript based on the by the reviewer’s suggestions.

This manuscript has been edited by Mark Roche on May 18, 2021 and is considered to be improved in grammar, punctuation, spelling, verb usage, sentence structure, conciseness, general readability, writing style, and native English usage to the best of the editor's ability.

Your Sincerely,

The Authors

Reviewer 1:

Dear Authors,

The paper aims to analyzed the percentage of caries and oral disease among children with special needs.

the article is very interesting but I suggest a revision.

Line 46 " According to Healthy People 2010 in the USA, " this should be a citation? if yes please provide it

  • We are grateful for the helpful suggestion. The authors corrected the citation of the sentence in Line 46-49. (Please see References in Line 375-376)

Line 55-57 please rewrite the sentence it is a little bit confusing.

  • We are thankful for the suggestion in order to make the manuscript more fluent for readers, we have revised the sentence related secondary condition in Line 54-58.

Line 84 please provide the mean age

  • We appreciate the suggestion The authors added the mean age of the population in Line 84-85.

Line 105 what type of mirror, what type of dental explore please provide brand and type for each.

  • We are grateful for the helpful suggestions. Authors added the brand and type of the dental mirror and dental explorer in Line 108-109.

Line 111 please add names abbreviation of the six trained and the "gold standard" (S.S)

  • We are thankful for the suggestion The authors have added names abbreviation of the six trained and the gold standard in Line 113-115.

Line 117 please explain the abbreviation  "evaluated using dmt + DMT and dms + DMS indices to "

  • We appreciate the suggestion The authors descripted the related abbreviation of dmt + DMT and dms + DMS in Line 121-126.

Line 123 "CPI" add brand and type

  • We are grateful for the helpful suggestions. Authors added the brand and type of the CPI probe in Line 127.

Line 169 please use the already cited abbreviations.

"The children with mild/moderate disabilities or with hearing or voice disability had a statistically significantly higher decayed teeth and surfaces need for dental treatment than those with the other severity of disability 171 or classification of disabilities "

Mild-moderate disabilities (MD)?

hearing or voice disability  (HVD)?

  • We are thankful for the suggestion In this manuscript, the authors had defined MD as the children with multiple disabilities in Line 95, not the children with mild/moderate disabilities (Line 176). The authors changed the phrase “hearing or voice disability” to abbreviations “HVD” in Line 177.

Please rewrite the discussion 

There are several repetitions about the same topic (e.g. "proper sweet habits and inadequate tooth-brushing behavior" 

"diet control or food habits play a critical role in the causation of dental caries" 

"Most TNs originate from dental caries, which is associated with oral care habits and CD’s tooth-brushing behavior"

"If parents have higher occupation level, we inferred that the parents 295 will have sufficient financial capacity to take care of their children's oral health "

please check the discussion too long and repetitive.

  • We appreciate the helpful suggestion The authors have rewritten the discussion of this manuscript. (Please see Discussion in Line 220-322)

In the limitations, you should add that the observation was assessed outside the dental clinic with limited accessibility and difficulties for the patients handling.

  • We are grateful for the helpful suggestion. We added “The observation was assessed outside the dental clinic with limited accessibility and difficulties for the patients handling.” in Line 315-316.

Reviewer 2 Report

I would like to congratulate the topic addressed, since it is not always an easy area to deal with. However, I would like clarification on some points:

Line 74 – not always is possible to use the nitrous oxide/oxygen sedation since it obliges the acceptance of the mask and breath by the nose. Oral sedation, is it considered?

Line 94 -What do you consider as intellectual disability, and multiple disabilities. It should be clarified and if possible, give some examples

Line 122 - The Plaque Index assessment seems questionable. First, the authors refer that 6 indicative surfaces, however, it seems that they meant to state 6 teeth, right? Otherwise, 6 surfaces from an overall mouth are expectedly not enough, and highly susceptible to data bias. But then later the authors refer to the teeth but mixed information on tooth surfaces. Please clarify this.

Line 128 - Also, do you mean that PI was assessed through a colourimetric method? And how it is with some cases that cannot rinse?

Line 257 - GA is expensive and complicated.  Complicated how come, since if facilitates the dental treatment

In the discussion is not addressed Plaque Index.

Thank once again for t

Author Response

Author's Response to Decision Letter for (Manuscript ID: jpm-1206711)

Title: Dental treatment needs and related risk factors among school children with special needs in Taiwan

Dear Reviewer,

The authors sincerely appreciate for the opportunity to consider our manuscript for publication, subject to major amendments. We have revised this manuscript based on the by the reviewer’s suggestions.

This manuscript has been edited by Mark Roche on May 18, 2021 and is considered to be improved in grammar, punctuation, spelling, verb usage, sentence structure, conciseness, general readability, writing style, and native English usage to the best of the editor's ability.

Your Sincerely,

The Authors

Reviewer 2:

Comments and Suggestions for Authors

I would like to congratulate the topic addressed, since it is not always an easy area to deal with. However, I would like clarification on some points:

Line 74 – not always is possible to use the nitrous oxide/oxygen sedation since it obliges the acceptance of the mask and breath by the nose. Oral sedation, is it considered?

  • We are grateful for the helpful suggestions. The authors strongly agreed that oral sedation is indeed another possible consideration. Although the patient is less able to wear a nasal mask and reduce the inhalation and effectiveness of nitrous oxide/oxygen sedation. However, after training and communication, some patients can be used the nitrous oxide/oxygen sedation, and reached the expected effectiveness. Oral sedation is not suitable for patients who need long-term treatment because of the large difference between individuals, the effect is unpredictable, and less be used in Taiwan. But we added the oral sedation in introduction in Line 75.

Line 94 -What do you consider as intellectual disability, and multiple disabilities. It should be clarified and if possible, give some examples

  • We are thankful for the helpful suggestion. The CD were categorized into intellectual disability and/or multiple disabilities according to their disability identification certificates and described in Line 84-97.

Line 122 - The Plaque Index assessment seems questionable. First, the authors refer that 6 indicative surfaces, however, it seems that they meant to state 6 teeth, right? Otherwise, 6 surfaces from an overall mouth are expectedly not enough, and highly susceptible to data bias. But then later the authors refer to the teeth but mixed information on tooth surfaces. Please clarify this.

  • We appreciate the suggestion It is true that PI were scored form 6 indicative surfaces. These 6 indicative surfaces are the surfaces of abundant plaque from 6 indicative teeth. The authors had revised the sentences in Line 126-128.

Line 128 - Also, do you mean that PI was assessed through a colourimetric method? And how it is with some cases that cannot rinse?

  • We are grateful for the helpful suggestions. PI was not assessed through a colourimetric method (with disclosing agents) in this study. Dental plaque biofilm accumulates on the surface of teeth and can be visible with naked eye and an explorer. Therefore, the participants did not need to rinse. The method often be used to assess PI in oral hygiene researches. The authors revised the sentence of PI evaluation in Line 127.

Line 257 - GA is expensive and complicated. Complicated how come, since if facilitates the dental treatment

  • We are thankful for the helpful suggestions. We agreed the opinion of the reviewer. After discussed, we decided to delete the unnecessary sentences of the manuscript in Line 263-264.

In the discussion is not addressed Plaque Index.

  • We appreciate the suggestion We added a paragraph to discuss plaque in Line 240-256.

Round 2

Reviewer 2 Report

Thank for the clarifications, however it is still not enough clear the The methods:

Line 126 (6 indicative surfaces - 4 buccal and 2 lingual surfaces) - I do not understand which 4 bucal, and 2 lingual surfaces  are.

I would like some explanation

Author Response

Dear Reviewer:​

Thank you for your comment concerning our manuscript. This comment is valuable and very helpful for revising and improving our paper, as well as the important guiding significance to our researches. We have studied comments carefully and have made correction which we hope meet with approval.

Your Sincerely,

The Authors

Reviewer 2:

Thank for the clarifications, however it is still not enough clear the methods:

Line 126 (6 indicative surfaces - 4 buccal and 2 lingual surfaces) - I do not understand which 4 buccal, and 2 lingual surfaces are.

I would like some explanation

  • We appreciate the comment. We have made correction to make the sentence easier for readers to understand in Line 126-127.
